# Electrochemical ELISA Protein Biosensing in Undiluted Serum Using a Polypyrrole-Based Platform

**DOI:** 10.3390/s20102857

**Published:** 2020-05-18

**Authors:** Sunil K. Arya, Pedro Estrela

**Affiliations:** 1Department of Electronic & Electrical Engineering, University of Bath, Claverton Down, Bath BA2 7AY, UK; sunilarya333@gmail.com; 2Centre for Biosensors, Bioelectronics and Biodevices (C3Bio), University of Bath, Claverton Down, Bath BA2 7AY, UK

**Keywords:** carboxyl polypyrrole, electrochemical ELISA, TNF-α, polymeric alkaline phosphatase, biomarker, differential pulse voltammetry, biosensor

## Abstract

An electrochemical enzyme-linked immunosorbent assay (ELISA) biosensor platform using electrochemically prepared ~11 nm thick carboxylic functionalized popypyrrole film has been developed for bio-analyte measurement in undiluted serum. Carboxyl polypyrrole (PPy-COOH) film using 3-carboxy-pyrrol monomer onto comb-shaped gold electrode microarray (Au) was prepared via cyclic voltammetry (CV). The prepared Au/PPy-COOH was then utilized for electrochemical ELISA platform development by immobilizing analyte-specific antibodies. Tumor necrosis factor-alpha (TNF-α) was selected as a model analyte and detected in undiluted serum. For enhanced performance, the use of a polymeric alkaline phosphatase tag was investigated for the electrochemical ELISA. The developed platform was characterized at each step of fabrication using CV, electrochemical impedance spectroscopy and atomic force microscopy. The bioelectrodes exhibited linearity for TNF-α in the 100 pg/mL–100 ng/mL range when measured in spiked serum, with limit of detection of 78 pg/mL. The sensor showed insignificant signal disturbance from serum proteins and other biologically important proteins. The developed platform was found to be fast and specific and can be applicable for testing and measuring various biologically important protein markers in real samples.

## 1. Introduction

The development of reliable, fast and cost-effective methods for biomarker analysis has gained much needed attention for improved healthcare. The fast and accurate estimation of health biomarkers using such methods at the early stages of diseases can provide good prognosis and better chances for effective therapies [1,2,3]. Research in the biomedical diagnosis field has developed various methods and devices based on e.g., optical, calorimetric and electrochemical techniques for such estimation of biomarkers [4,5,6,7,8,9]. Among various techniques and methods, to date the use of the optical enzyme-linked immunosorbent assay (ELISA) is considered as gold standard and used in real sample testing [10]. In most cases, optical ELISA provides accurate and reliable data for biomarker detection; however, it requires complex equipment, long timings and skilled workers for testing and maintenance. Therefore, there is a demand for other simpler technologies to achieve high quality results in a shorter time and with easier testing procedures. Recently, electrochemical immunosensors, exhibiting high sensitivity, accuracy, simplicity and low cost, have received much attention [3,10,11]. Electrochemical ELISA, which combines a sandwich ELISA approach with electrochemical detection, is a promising technique that can provide high specificity due to the use of a sandwich assay and high sensitivity due to use of improved detection tags and electrochemical measurement [12]. However, to develop better electrochemical ELISA tests that can replace optical ELISAs in healthcare systems, significant research is still required and necessitate further collaborative efforts of researchers in various fields. In one such area of improvement, research is required in finding suitable matrices for immobilization of biomolecules, where the capturing molecule remains stable and active after immobilization, which is crucial for sensor development and one of the deciding factors in stability of developed sensor [13,14].

As a matrix for biomolecule immobilization, conjugated conductive polymers (CP), owing to their redox properties, electrical conductivity and compatibility with bioreceptor association, have attracted significant attention in the past decade in biosensor development [15,16]. Among various CPs, polypyrrole (PPy), known for its stability, conductivity and biocompatibility, has been the most intensively investigated and widely employed in biosensor applications. However, PPy suffers from the absence of functional groups and needs prior chemical modification to generate functional chemical groups such as carboxylic acids or amines for the stable immobilization of biomolecules [16]. Such chemical modification is time consuming and can affect the PPy’s chemical structure and hinder the conductivity of its backbone. In order to avoid a chemical modification step, in the present study we have developed an electrochemical ELISA sensor platform employing 3-carboxy pyrrole (Py-COOH) for polymerization and covalent binding of antibody using the carboxyl groups of polymer. PPy-COOH has been used previously for electrochemically preparing nanothin films for electrochemical-transmission surface plasmon resonance spectroscopy studies and for making biosensors for the detection of human immunoglobulin G (IgG) [17,18]. Also, its copolymer with pyrrole has been utilized for developing label-free impedimetric biosensors for *Salmonella Typhimurium* [16]. To the best of our knowledge, this is the first report on using nanothin films of PPy-COOH for developing electrochemical ELISA-based immunosensors.

Tumor necrosis factor-α protein (TNF-α) is a 157 amino acid long polypeptide forming a 47–55 kDa trimer cytokine [19,20]. It has been reported to play a crucial role in various immune and inflammatory progressions [21,22]. TNF-α exists in picogram per milliliter levels in the blood of healthy humans [23,24]. Studies by Milani et al. showed the TNF-α level in normal subjects to be 0.89 ± 0.40 pg/mL; range, 0.5 to 9.7 pg/mL [24]. However, it has been reported to increase by ten- to hundred-fold in the case of pathological conditions like rheumatoid arthritis (RA), thus making it a useful inflammatory biomarker [21,22,25,26]. Furthermore, TNF-α has been involved in many diseases like Crohn’s disease, neurodegenerative diseases (Alzheimer’s, Parkinsons), rejection to clinical transplantation, sepsis, and cancer, amongst others [27,28,29,30,31,32]. Apart from these diseases, it has also been shown to play negative role in wound healing [33]. At present, TNF-α is determined using techniques such as optical ELISA, radioimmunoassay and time resolve fluorescence assay [34,35,36]. Such techniques provide precise estimation; however, they require complex, expensive equipment that can only be run in central laboratories by skilled professionals.

In the present study, the advantages of carboxyl functionalized pyrrole (Py-COOH), electrochemical ELISA and polymeric alkaline phosphatase (PALP) have been combined to achieve enhanced high sensitivity with reliable estimation of TNF-α in spiked serum. PPy-COOH modified comb-shaped gold microelectrodes have been used to covalently immobilize monoclonal TNF-α antibody. Non-specific binding and false signals from detection in serum were avoided by the use of a blocker containing proprietary proteins in phosphate buffer with Tween20. Furthermore, the use of PALP gives higher loading of enzyme during binding, resulting in enhanced production of 4-aminophenol from 4-aminophenyl phosphate (4-APP) biocatalyzation, thus giving an improved redox signal.

## 2. Materials and Procedures for Development and Testing

### 2.1. Chemicals

Primary monoclonal antibody (product code: 502802) and secondary monoclonal antibody (product code: 502904) with attached biotin and TNF-α (product code: 570104) were purchased from BioLegend (San Diego, CA, USA). Alkaline phosphatase (ALP) and polymeric alkaline phosphatase (PALP) with conjugated streptavidin (enhanced Strept-AP) (product code: 5150N) was purchased from Kem-En-Tec Diagnostics (Taastrup, Denmark). Human serum (product code: H4522-100 mL) from human male AB plasma was purchased from Sigma (Gillingham, UK). Starting blockers TBST20 (SB-TBST) and PBST20 (SB) were purchased from Fisher Scientific (Loughborough, UK). 4-APP was purchased from Santa Cruz Biotechnology (Dallas, TX, USA). Pyrrole-3-carboxylic acid and lithium perchlorate were purchased from Sigma and Femto TBST was obtained from G-Biosciences (St. Louis, MO, USA). The rest of the chemicals and reagents used were of analytical grade and used without modification.

### 2.2. Apparatus

Electrochemical impedance spectroscopy (EIS) and cyclic voltammetry (CV) measurements were performed using a µAutolab III/FRA2 potentiostat/galvanostat (Metrohm, Netherlands) running on NOVA software. For EIS measurements, separate gold electrodes were used as counter and pseudo reference electrodes, respectively, and the developed electrode at each step was used as the working electrode. Measurements were performed in the 100 kHz–100 mHz frequency range, at applied AC amplitude of 25 mV and open circuit potential, i.e., equilibrium potential existed between electrodes incubated in test solution, without external biasing. Electrode development was also characterized via cyclic voltammetry (CV). CV and EIS studies were performed in 0.1 M KCl (50 μL) containing 5 mM [Fe(CN)_6_]^3*−*/4*−*^ as a redox probe. PPy-COOH deposition was also characterized using atomic force microscopy (AFM) imaging in ambient contact mode and scanning electron microscopy. AFM investigations were carried out via MultiMode NanoScope (Bruker, Germany). The AFM system utilized version 6 software with IIIa controller. For imaging, electrodes were scanned using 10 nm diameter AFM ContAl-G tips (BudgetSensors, Bulgaria), followed by image processing using Bruker’s version 1.5 NanoScope Analysis software. Scanning electrode microscopy (SEM) studies were performed to confirm selective PPY-COOH deposition on the electrodes using a JEOL JSM-6480 SEM (JEOL, Peabody, MA, USA). For TNF-α detection and estimation, differential pulse voltammetry (DPV) measurements were performed at −0.2–0.45 V under the applied 25 mV amplitude, 3 mV step potential, 0.03 s interval time, 0.02 s modulation time and 100 mV/s scan rate.

### 2.3. Gold Electrode Preparation

Counter and pseudo reference electrodes, along with interdigitated working electrodes of gold, were fabricated using lithography and micro-fabrication processes on silicon oxide/silicon substrates [37]. In sensor development, one side of the interdigitated electrode possessing 3200 μm long and 5 μm wide electrodes at a 25 μm gap, spread over a 5500 μm length, was used as the working electrode. Fabricated electrodes were cleaned using ethanol, acetone and deionized H_2_O followed by UV-ozone treatment for half an hour before utilization.

### 2.4. PPy-COOH Deposition and Anti-TNF-α Immobilization

Before PPy-COOH deposition onto fresh clean gold electrodes, a chamber covering the interdigitated area was made using laser cut tape, capable of holding 50 to 100 µL of solution (Figure 1). For electrochemical deposition, 50 µL of 50 mM pyrrole-COOH in 0.5 M LiClO_4_ in H_2_O was poured into the chamber and PPy-COOH was electrochemically deposited via sweeping the voltage at 50 mV/s between −0.1 and 0.8 V. Five voltage sweeps were applied through cyclic voltammetry for optimum film deposition. After electrochemical deposition, the PPy-COOH film was washed with miliQ water and dried by gentle air blow. Before anti-TNF-α immobilization, the carboxylic groups on the PPy-COOH film were activated by incubating the PPy-COOH modified surface with 50 µL of EDC (0.4 M)/NHS (0.1 M) in H_2_O for 30 min. After activation, the electrode was washed with H_2_O and then with PBS (10 mM, pH 7.4). Extra PBS was removed from the electrode surface and treated with 50 µL of anti-TNF-α (10 µg/mL) in PBS for 90 min at room temperature. The antibody modified surface was then rinsed with PBST and PBS followed by incubation in 50 µL SB for 1 h to block empty sites and to prevent nonspecific binding. After incubation, the extra SB was decanted and the prepared Au/PPy-COOH/anti-TNF-α/SB electrodes were kept at 4 °C for storage. Figure 1b illustrates the steps involved in Au/PPy-COOH/anti-TNF-α/SB bioelectrode fabrication and immunoassay.

### 2.5. TNF-α Binding and Estimation Studies

For TNF-α binding and estimation, Au/PPy-COOH/anti-TNF-α/SB bioelectrodes were treated with the chosen concentrations of TNF-α spiked in serum. The electrodes were tested for incubation times of 10, 20, 30 and 40 min for optimization. Optimum response was observed when electrodes were treated with antigen, secondary antibody and enzyme tag for 30 min each and then incubated with enzyme substrate for 20 min before the electrochemical measurement. Thus, 30 min incubations for the initial 3 steps and 20 min incubation for last step were used in this study for the sensing. For the immunoassay, the following steps were performed at room temperature (22 ± 1 °C) in a sequential manner:Antigen incubation: 30 min.Washing using washing buffer (Femto TBST, 1x): three times.Interaction with biotin modified secondary antibody solution (50 µL, 10 µg/mL) in SB-TBST: 30 min.Washing using washing buffer (Femto TBST, 1x): three times.Interaction with streptavidin modified alkaline phosphatase (ALP) or polymeric alkaline phosphatase (PALP) solution (50 µL, 10 µg/mL) in SB-TBST: 30 min.Washing using washing buffer (Femto TBST, 1x): three times.Final interaction with 50 µL of 2 mg/mL 4-APP in de-aerated tris-HCl buffer (100 mM, pH 9.0 containing 4 mg/mL MgCl_2_·6H_2_O): 20 min.Electrochemical oxidation signal recording using differential pulse voltammetry.

The developed Au/PPy-COOH/anti-TNF-α/SB bioelectrode was also tested for protein interference studies in non-spiked serum and in serum samples spiked with human epidermal growth factor receptor 2 (HER2), C-reactive protein (CRP) and prostate-specific antigen (PSA). Each experiment was carried out in triplicate on three separate electrodes and shown using error bars.

## 3. Results and Discussion

### 3.1. EIS and CV Investigations

The Au/PPy-COOH/anti-TNF-α/SB bioelectrode fabrication was characterized at each step using CV and EIS (Figure 2). After PPy-COOH deposition, a slight decrease in current and slight increase in charge transfer resistance (*R*_ct_) was observed (Figure 2a,b). Such small but significant changes indicated the deposition of a very thin PPy-COOH film of around 11 nm, as confirmed via the AFM study. Scan rate studies (Figure 2c,d) further suggest that polymer deposition does not affect significantly the conducting properties of the gold electrode and exhibit similar redox behavior as showed by the bare gold electrode. Furthermore, in scan rate studies between 30 and 100 mV/s using CV, electrode showed linear variation in peak current with a square root of scan rate (Figure 2e,f), indicating a diffusion-controlled process [38]. CV and EIS studies and comparison of the electrodes with and without PPy-COOH on adjacent electrodes on the same chip (Figure 2g,h) suggest that PPy-COOH deposition occurs selectively on the desired electrode, while the other electrode without PPy-COOH shows CV and EIS responses similar to the electrode just kept immersed in polymerization solution without applying any voltage or CV scans. The lower current response and higher *R*_ct_ in the non-modified electrode kept immersed in polymerization solution compared to the fresh clean blank electrode may be attributed to the adsorption of non-conducting molecules of pyrrole on the fresh clean gold surface. The difference in electrode with and without PPy-COOH on the same chip is also clearly visible in SEM images (Figure 3a).

### 3.2. SEM and AFM Characterization for PPy-COOH Deposition

PPy-COOH deposition was also confirmed using SEM and AFM studies. Figure 3a shows the SEM image of the electrode with PPy-COOH deposited on one side of interdigitated electrode, which was used as the working electrode. The dark color on one side of interdigitated electrode suggested successful deposition of PPy-COOH, whereas the clear shining electrode suggested no deposition on PPy-COOH on the other pair of interdigitated electrodes. AFM studies were further used to confirm PPy-COOH deposition and to measure the thickness of the deposited polymer. A clear change in morphology from spiky in un-modified gold electrode (Figure 3c) to globular after deposition of PPy-COOH (Figure 3d) further confirmed the successful deposition of PPy-COOH. Further, AFM scanning was used to measure the height of two sides of electrode fingers with and without PPy-COOH on the same chip and found to be around 11 nm via step analysis (Figure 3b).

### 3.3. Differential Pulse Voltammetric Studies for TNF-α Detection

Different concentrations of TNF-α (*c*_TNF-α_) between 100 pg/mL and 100 ng/mL in pure serum were investigated with the Au/PPy-COOH/anti-TNF-α/SB bioelectrode via DPV (Figure 4a). Analysis of Figure 4a revealed that the sensor is insensitive to serum proteins, as no response was observed for serum. Also, the Au/PPy-COOH/anti-TNF-α/SB bioelectrode showed no signal up to 1 ng/mL of TNF-α, indicating no sensitivity for such low concentrations. For concentrations of 5 ng/mL and above, the peak current was found to increase with the increase in concentration, indicating successful bioassay and ability of the Au/PPy-COOH/anti-TNF-α/SB bioelectrode for TNF-α detection in serum. The increasing peak current with *c*_TNF-α_ corresponds to the increasing concentration of 4-amino phenol produced in the biocatalyzation of 4-APP by captured ALP-streptavidin during bioassay. With higher production of 4-amino phenol, a higher electrochemical oxidation current was observed during the DPV scans. The enlarged image shown in Figure 4b reveals a distinctive signal from 5 ng/mL. Further, analysis shown in Figure 4c suggests a linear behavior of the Au/PPy-COOH/anti-TNF-α/SB bioelectrode for TNF-α detection, which can be represented as *I* (µA) = 0.38 µA + 0.054 µA/(ng/mL) *c*_TNF-α_ (ng/mL). Moreover, a sensitivity of 54 nA/(ng/mL) and limit of detection of 85 pg/mL (3xSD of blank/sensitivity) were observed for the Au/PPy-COOH/anti-TNF-α/SB electrode. The 5% error observed in the triplicated set of experiments shown via the error bars indicated acceptable repeatability. The results of these studies using the Au/PPy-COOH/anti-TNF-α/SB electrode indicate the suitability of using PPy-COOH modified gold surface for the development of a new biosensor platform for the detection and estimation of biologically relevant protein biomarkers in real samples.

To improve the observed characteristics for the developed sensor in a lower concentration range, further studies were made using polymeric APL. Figure 5a shows the DPV spectra for spiked serum with TNF-α in the 100 pg/mL–100 ng/mL range using the Au/PPy-COOH/anti-TNF-α/SB sensor, when tested with PALP. The results clearly indicate more than seven times higher sensitivity and current response than with standard ALP. Furthermore, the DPV spectra for lower concentrations (Figure 5b) indicated that TNF-α detection using PALP results in a clear and distinguishable response, even at 100 pg/mL. Even though the use of PALP resulted in a higher response to un-spiked serum, the response for 100 pg/mL was much higher and clearly distinctive. Better and higher response indicated the presence of a larger number of ALP molecules per antigen captured during antibody–antigen interactions. Further analysis shown in Figure 5c suggests enhanced linear behavior of the sensors towards TNF-α detection, which can be represented as *I* (µA) = 1.44 µA + 0.383 µA/(ng/mL) *c*_TNF-α_ (ng/mL). Moreover, sensitivity of 383 nA/(ng/mL) and limit of detection of 78 pg/mL was obtained in this case. The 5% error observed in triplicated set of experiment shown via error bars indicates acceptable reproducibility and consistency of the electrodes prepared. Furthermore, the sensors were stable for periods of up to 4 weeks when stored at 4 °C. The enhanced characteristics observed with the use of PALP for TNF-α estimation in spiked serum clearly suggest that the PALP approach provides the opportunity of using PPy-COOH-modified gold surface for the development of a new biosensor platform with enhanced and sensitive detection and estimation of biologically relevant protein biomarkers in real samples.

### 3.4. Interference Studies from Other Proteins

The response of the sensor towards proteins in serum as well as towards protein biomarkers such as HER2, CRP and PSA at 100 ng/mL in spiked serum was studied in order to investigate interference effects. Figure 6 shows the DPV responses of the Au/PPy-COOH/anti-TNF-α/SB bioelectrode for various possible interferents: non-spiked serum; 100 ng/mL of HER2, CRP and PSA in serum. The results clearly show the selectivity of the sensor for TNF-α, with negligible response to HER2, CRP, PSA or proteins in serum. Thus, the Au/PPy-COOH/anti-TNF-α/SB bioelectrode offered a viable platform to estimate protein markers in real samples. Table 1 compares the results of the present study with other reported results for TNF-α, clearly suggesting that the Au/PPy-COOH/anti-TNF-α/SB bioelectrode can be used to get a better response in a wide dynamic range in undiluted serum samples.

## 4. Conclusions

In summary, the PPy-COOH matrix-based system with PALP tag-based enhancement strategy offered a novel electrochemical ELISA biosensor system to measure protein biomarkers in undiluted serum in a selective and specific manner with enhanced sensitivity. The Au/PPy-COOH/anti-TNF-α/SB bioelectrode with PALP conjugated reagents showed linear response for TNF-α estimation in the range 100 pg/mL–100 ng/mL. The developed system showed a sensitivity of 383 nA/(ng/mL) and a limit of detection of 78 pg/mL. The sensor further exhibited selectivity for TNF-α with negligible signals for HER2, CRP, PSA and other serum protein. The results of the present study clearly suggest that the PALP approach with a PPy-COOH-based electrochemical ELISA platform provides a new opportunity for enhanced and sensitive detection and estimation of protein biomarkers in serum. With the use of suitable antibodies, the technique can be expanded to the detection of a wide range of clinically relevant biomarkers in real samples. The electrochemical detection allows for miniaturization and multiplexing in low-cost systems.

## Figures and Tables

**Figure 1 sensors-20-02857-f001:**
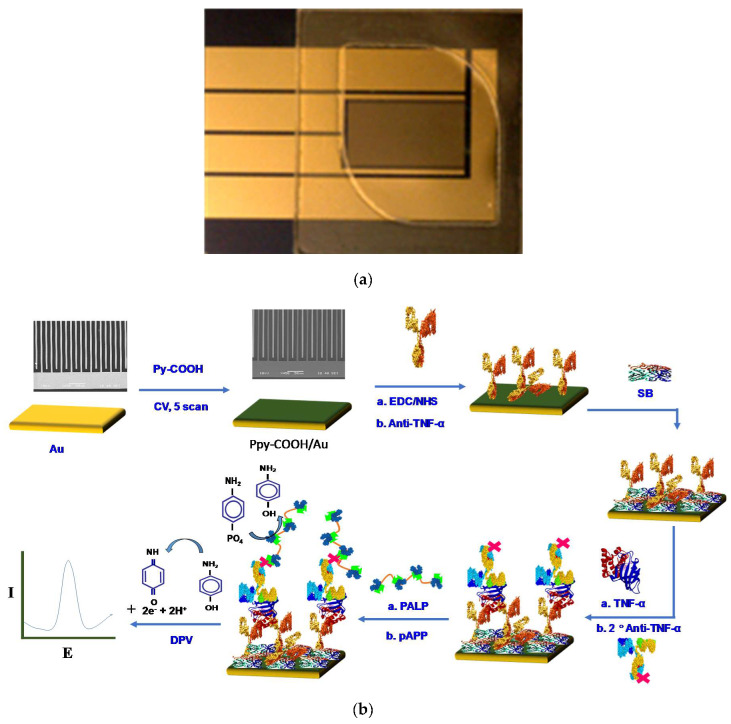
(**a**) Optical image of fabricated chip with laser cut chamber; (**b**) schematic for Au/PPy-COOH/anti-TNF-α/SB electrode fabrication and immunoassay.

**Figure 2 sensors-20-02857-f002:**
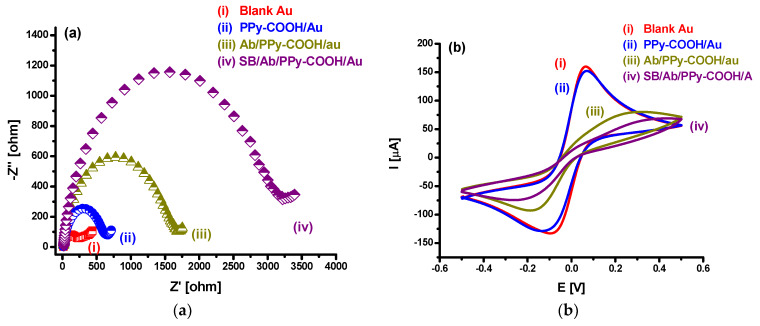
Characterization of Au/PPy-COOH/anti-TNF-α/SB electrode fabrication using (**a**) electrochemical impedance spectroscopy (EIS) and (**b**) cyclic voltammetry (CV). Scan rate study of (**c**) blank Au, (**d**) PPy-COOH/Au. (**e**,**f**) show the changes in peak current with square root of scan rate for (**c**,**d**), respectively. Study of electrode incubation effect in polymerization solution via (**g**) EIS and (**h**) CV.

**Figure 3 sensors-20-02857-f003:**
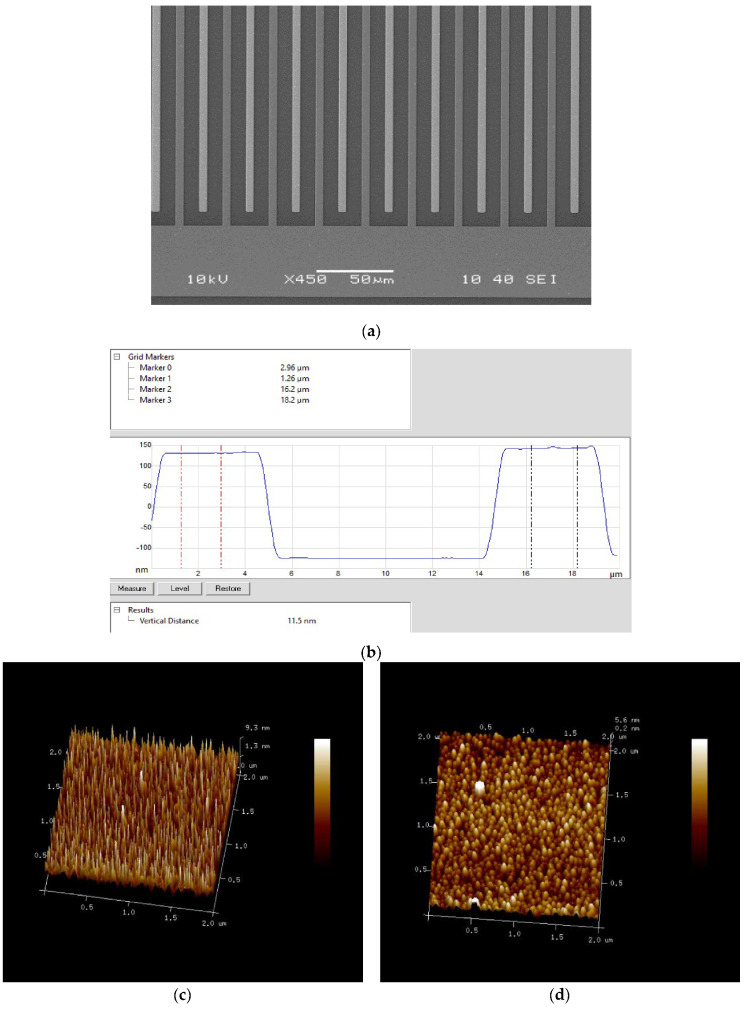
(**a**) SEM characterization image of PPy-COOH deposition; (**b**) thickness of PPy-COOH deposition via AFM study. (**c**) AFM image of blank Au and (**d**) PPy-COOH/Au.

**Figure 4 sensors-20-02857-f004:**
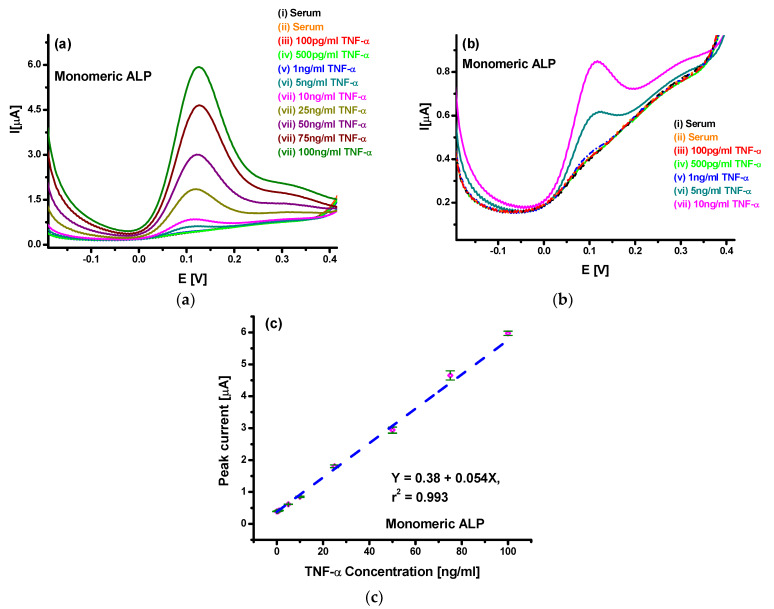
(**a**) Differential pulse voltammetry (DPV) studies for TNF-α spiked serum in the 100 pg/mL–100 ng/mL range using the Au/PPy-COOH/anti-TNF-α/SB bioelectrode; (**b**) DPV studies for the Au/PPy-COOH/anti-TNF-α/SB electrode towards TNF-α concentration up to 10 ng/mL; (**c**) Linear range analysis of the Au/PPy-COOH/anti-TNF-α/SB electrode towards TNF-α concentration.

**Figure 5 sensors-20-02857-f005:**
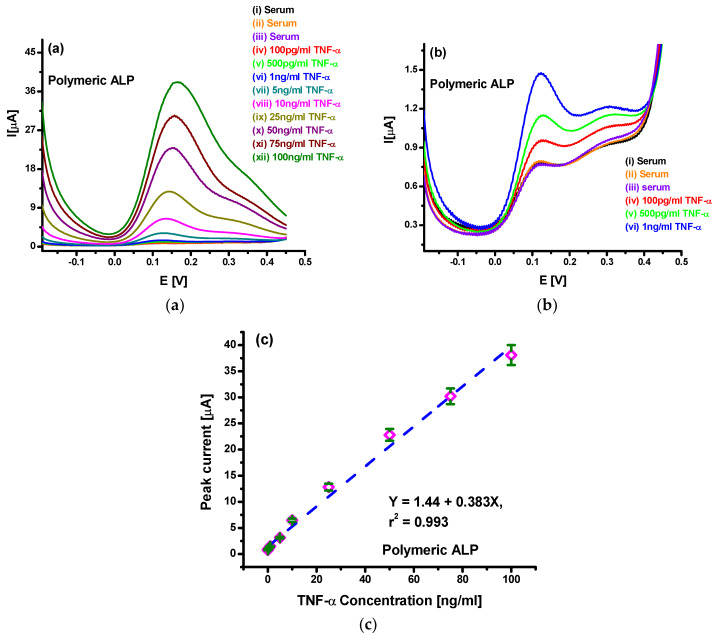
(**a**) DPV studies for TNF-α spiked serum in the 100 pg/mL–100 ng/mL range using Au/PPy-COOH/anti-TNF-α/SB electrode with PALP; (**b**) DPV studies for the Au/PPy-COOH/anti-TNF-α/SB electrode towards TNF-α concentration up to 1 ng/mL; (**c**) Linear range analysis of the Au/PPy-COOH/anti-TNF-α/SB electrode towards TNF-α concentration.

**Figure 6 sensors-20-02857-f006:**
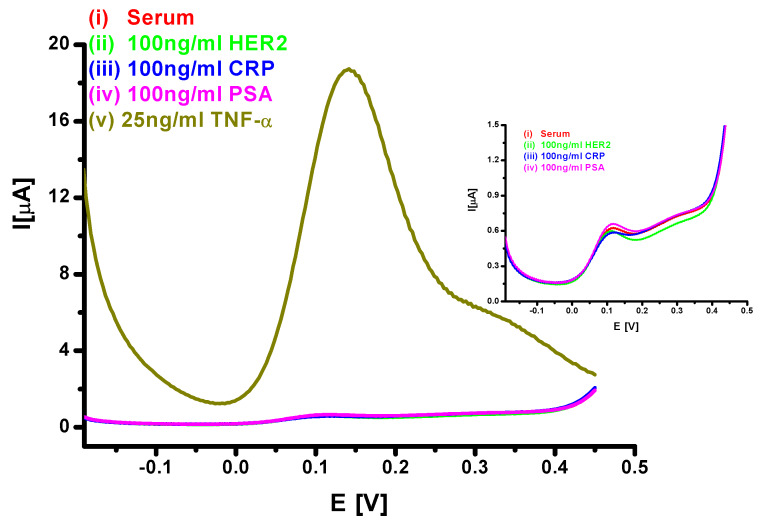
DPV response of the Au/PPy-COOH/anti-TNF-α/SB bioelectrode for various interferents in serum: (i) non-spiked serum; (ii) 100 ng/mL of HER2; (iii) 100 ng/mL of CRP; (iv) 100 ng/mL of PSA; and (v) 25 ng/mL of TNF-α for comparison.

**Table 1 sensors-20-02857-t001:** Characteristics of the the Au/PPy-COOH/anti-TNF-α/SB bioelectrode and those reported in the literature.

Sensor	Sample Medium	Detection Method	Linearity (ng/mL)	Detection Limit (ng/mL)	Ref.
SB/EA/anti-TNF-α/DTSP SAM/Au	Undiluted serum	DPV	0.5–100	0.06	[4]
MB/TNF-α aptamer SAM/ Au	Cell culture medium	SWV	9–88	5.46	[39]
BSA/anti-TNF-α/MUA SAM/Au	Buffer	QCM	40–2000	25	[40]
BSA/anti-TNF-α/DNA nanostructure/Au	1% BSA in PBS	Amperometry	0.1−2.5	0.1	[41]
SB/anti-TNF-α/PPy-COOH/Au	Undiluted serum	DPV	0.1–100	0.078	present work

DPV: differential pulse voltammetry; DTSP: dithiobis(succinimidyl propionate); EA: ethanolamine; MB: methylene blue; MUA: mercaptoundecanoic acid; QCM: quartz crystal microbalance; SAM: self-assembled monolayer; SWV: square wave voltammetry

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
