# Peer review of "Electrochemical ELISA Protein Biosensing in Undiluted Serum Using a Polypyrrole-Based Platform"

_sensors, 2020, doi:10.3390/s20102857_

Round 1

Reviewer 1 Report

This work presented an electrochemical ELISA platform composed of the gold electrode microarray functionalized by carboxyl polypyrrole. Enhanced sensitivity and reduced non-specific binding could be achieved by the combined use of carboxyl polypyrrole and polymeric alkaline phosphatase. The work is scientifically sound and would be generally interesting to Sensors readers. Therefore, this paper is recommended for publication after the authors have addressed the concerns outlined below.

  1. Please recheck the figures carefully. Take some for examples: the legends of Figure 2 (a) and (b), the curve colors of Figure 2 (e) and (f). In Figure 6, 25mg/ml should be 25 ng/mL.
  2. How many measurements could be implemented on one interdigitated electrode array?
  3. A missing part is a comparison of the present method with other representative electrochemical methods for the detection of TNF-a.

Author Response

Reviewer #1

This work presented an electrochemical ELISA platform composed of the gold electrode microarray functionalized by carboxyl polypyrrole. Enhanced sensitivity and reduced non-specific binding could be achieved by the combined use of carboxyl polypyrrole and polymeric alkaline phosphatase. The work is scientifically sound and would be generally interesting to Sensors readers. Therefore, this paper is recommended for publication after the authors have addressed the concerns outlined below.

Response: Thank you for this appreciation.

  1. Please recheck the figures carefully. Take some for examples: the legends of Figure 2 (a) and (b), the curve colors of Figure 2 (e) and (f). In Figure 6, 25mg/ml should be 25 ng/mL.

Response: Thank you for your comment. All figures have been checked and corrections have been incorporated in the revised manuscript.

  1. How many measurements could be implemented on one interdigitated electrode array?

Response: Each Au/PPy-COOH/anti-TNF-α/SB bioelectrode was utilized for one concentration testing only in order to ensure that we have reproducibility between electrodes.

  1. A missing part is a comparison of the present method with other representative electrochemical methods for the detection of TNF-a.

Response: Thank you for this suggestion. A comparison table has been added to the revised manuscript.

Reviewer 2 Report

The authors have developed an ELISA method for detection of a protein in serum based on electrochemical sensing. Although the methodology is not novel, the present work can be useful to the readers. However, in order to be considered for potential publication, the following comments should be addressed. My main comments are:   1) How does this paper compare with gold standard approaches? The authors should provide a comparison. Having a table would help.
  2) Moreover, the authors have not studied the stability/life time of their structure. Such evaluations should be added.   I have made some other minor comments (including formatting and grammatical edits) in the attached annotated pdf file. 

Author Response

Reviewer 2

1) How does this paper compare with gold standard approaches? The authors should provide a comparison. Having a table would help.

Response: Thank you for this suggestion. A comparison table has been added to the revised manuscript.

2) Moreover, the authors have not studied the stability/life time of their structure. Such evaluations should be added.  

Response: Although a systematic study on the shelf life of the sensors was not performed, a few of the prepared bioelectrodes were stored for 4 weeks at 4 °C without loosing their performance, presumably due to the stabilization of the surfaces with the StartingBlock. A short comment to this effect has been added to the manuscript.

I have made some other minor comments (including formatting and grammatical edits) in the attached annotated pdf file. 

Page 1 line 21: How does it compare with clinically-relevant concentrations that need to be detected?

Response: As mentioned in the introduction section “TNF-α exists in picogram per milliliter levels in the blood of healthy humans; however, it has been reported to increase by ten to hundred fold in the case of pathological conditions like rheumatoid arthritis (RA), thus making it a useful inflammatory biomarker.”

As concentration increases so much, the tested range is useful for clinical samples as the present sensor were used in undiluted serum. Also, the aim of work was to demonstrate PPy-COOH and PALP based platform for biomarker detection in electrochemical ELISA format, which can be applied to many other biological biomarkers like PSA. In any case we have added the clinical ranges of TNF-α for healthy humans.

Page 1, line 35 : “till date” Should be re-written, for example, "till this date" or "so far", etc.

Response: Changes have been incorporated in the revised manuscript.

Page 2, line 60: “3-carboxy pyrrole” Reference to other works using this compound and what application they were for can be useful to the readers. I'd suggest adding so.

Response: Changes with references have been incorporated in the revised manuscript.

Line63: What is the relevant clinical range for detection of TNF-\alpha? This should be provided.

Response: Text has been updated in the revised manuscript.

Line 85: Details about the serum sample need to be provided.

Response: Details have been added in the revised manuscript.

Line 173: Citation is required.

Response: Reference has been added in in the revised manuscript.

Line 182: In figures e and f, the curve colors are not consistent with the legends.

Response: Corrections have been made in the revised manuscript.

Line 183:The insets are too small and the texts are hardly readable.

Response: Changes have been made to improve the figure.

Line 198: The text size in Fig. b are too small and hardly readable.

Response: Figure has been improved for better clarity.

Line 216: The authors indicated the error bars are from three tests. It should be clarified whether they are three independent sets or just three data points measure using same sensor. Also, comment on sample-to-sample variation is needed.

Response: Three tests were made on three different sensors. Best possible precautions were taken to make identical solutions on different days to avoid sample to sample variation for same concentration. Even with possible slight variation in sample on day to day basis not much variation was observed in the sensor response as visible by the variation in error bars.

Line 222: "of" should be removed.

Response: Correction has been incorporated.

Line 223: and line 247: Figure (b) seems to be just a magnified version of (a) at low concentrations. To remove confusion, I would suggest showing (b) as inset of (a).

Response: Efforts were made to make fig b as an inset, however the text was not clear and unreadable. Also, signal clarity was not very good. Thus, figures were kept as they were originally.

Reviewer 3 Report

In this work, the authors reported an electrochemical ELISA for detection of protein biomarker using PPy-COOH modified interdigitated gold electrode and PALP. As a result, Au/PPy-COOH/anti-TNF-α/SB bioelectrode with PALP conjugated reagents showed about 7 times higher sensitivity and current response than with standard ALP. The manuscript is well-designed and easy to follow. But, there are some major issues that I feel need addressing.

1) What are the distinctive advantages of the obtained electrochemical ELISA? There are several references about ultrasensitive electrochemical immunosensors. In addition, compare the analytical performance of this sensing system with the other electrochemical immunosensors for TNF-α. 

2) There is no information about long term stability. How long can the bioelectrode be stored?

3) The authors utilized PBST20 (SB) as a blocking agent. Why did not the authors use BSA which is the most frequently used agent? Is there any special reasons to use the SB, except for cost effectiveness? Can the SB keep the blocking effect during or after a lot of washing steps?

4) Though anti-TNF- α could be covalently bonded with PPy-COOH by EDC/NHS coupling, antibody might be also detached passively onto the other pair of intergiditated gold electrode without PPy-COOH. Is there any effect by passively bonded antibody onto other gold electrode?

5) In general, activated –COOH by EDC/NHS was treated by quenching reagent to stop the further reaction. In this study, the authors did not use the quenching agent. Isn’t there some possibility for the remaining –COOH (activated) to bind with antigen directly?     

Minor points)

1) In Fig. 2 e,f: It is necessary to add the descriptions for e and f to figure caption. And legend for e and f should be corrected.

2) Please add the incubation temperature for each step into Section 2.1 TNF- α binding and estimation studies.

3) There are a few typos in manuscript. (for example, Fig. 6 : 25 mg/ml TNF-α … etc)   

Author Response

Reviewer 3

In this work, the authors reported an electrochemical ELISA for detection of protein biomarker using PPy-COOH modified interdigitated gold electrode and PALP. As a result, Au/PPy-COOH/anti-TNF-α/SB bioelectrode with PALP conjugated reagents showed about 7 times higher sensitivity and current response than with standard ALP. The manuscript is well-designed and easy to follow. But, there are some major issues that I feel need addressing.

Response: Thank you for this appreciation.

1) What are the distinctive advantages of the obtained electrochemical ELISA? There are several references about ultrasensitive electrochemical immunosensors. In addition, compare the analytical performance of this sensing system with the other electrochemical immunosensors for TNF-α. 

Response: There are various reports on ultrasensitive electrochemical immunosensors. However, most of them either report the measurement of analyte in buffer medium or in very diluted serum samples. In the present work using electrochemical ELISA we utilized the advantages of optical ELISA and electrochemical immunoassay and achieved sensitive detection in undiluted serum sample, which is very crucial for real world application. As suggested, a comparison table has been incorporated in revised draft.

2) There is no information about long term stability. How long can the bioelectrode be stored?

Response: Although a systematic study on the shelf life of the sensors was not performed, a few of the prepared bioelectrodes were stored for 4 weeks at 4 °C without loosing their performance, presumably due to the stabilization of the surfaces with the StartingBlock. A short comment to this effect has been added to the manuscript.

3) The authors utilized PBST20 (SB) as a blocking agent. Why did not the authors use BSA which is the most frequently used agent? Is there any special reasons to use the SB, except for cost effectiveness? Can the SB keep the blocking effect during or after a lot of washing steps?

Response: Both BSA and starting block (SB) were screened in initial investigations and SB was found superior in blocking effect even with lots of washing in various steps of the assay. Also, with SB, stored electrodes at 4ËšC were found more stable and active compared to BSA even after four weeks.

4) Though anti-TNF- α could be covalently bonded with PPy-COOH by EDC/NHS coupling, antibody might be also detached passively onto the other pair of intergiditated gold electrode without PPy-COOH. Is there any effect by passively bonded antibody onto other gold electrode?

Response: In theory, it is possible that some physically attached antibodies on neighbouring electrode might detached passively and result in variation. However, best efforts in washing were made to remove any physically adsorbed antibodies and low variation among various electrodes as shown by error bars suggest non-significant effect of detachment. To understand this better, more detailed investigations are required in the future.

5) In general, activated –COOH by EDC/NHS was treated by quenching reagent to stop the further reaction. In this study, the authors did not use the quenching agent. Isn’t there some possibility for the remaining –COOH (activated) to bind with antigen directly?    

Response: To avoid such possibility, 1hr blocking was performed using PBST 20 SB, which contain various sizes of proteins and effectively block such active COOH sites. 

Minor points)

  • In Fig. 2 e,f: It is necessary to add the descriptions for e and f to figure caption. And legend for e and f should be corrected.

Response: Changes have been incorporated in the revised manuscript.

  • Please add the incubation temperature for each step into Section 2.1 TNF- α binding and estimation studies.

Response: All the steps were carried out at room temperature. Text has been modified for clarity.

  • There are a few typos in manuscript. (for example, Fig. 6 : 25 mg/ml TNF-α … etc)   

Response: The manuscript has been further checked for typos.

Reviewer 4 Report

Authors reported the development of gold-based electrochemical immune-sensor for the detection of the antigen TNFa in intact human blood samples.

The electrode is coated with carboxy-polypyrrole and the whole system shows zero interference with intact blood.

In the plot A (Fig 4, page 8), two of the three DPV curves related to serum are completely covered by the 100 and 500 pg/mL response, while the same are repeated in the plot B on the right. Apparently, serum curves are not visible in plot A and information are redundant in plot B. Please change the color code (or the thickness) or delete the 100/500pg curves in plot A, in order to let the reader see the “zero response” with bare serum.

Please specify the exact product code (from Biolegend) of the primary antibody used to capture TNFa, and also if it is monoclonal or polyclonal.  

Author Response

Reviewer 4

In the plot A (Fig 4, page 8), two of the three DPV curves related to serum are completely covered by the 100 and 500 pg/mL response, while the same are repeated in the plot B on the right. Apparently, serum curves are not visible in plot A and information are redundant in plot B. Please change the color code (or the thickness) or delete the 100/500pg curves in plot A, in order to let the reader see the “zero response” with bare serum.

Response: The idea behind to include the curve for 100 and 500 pg/ml in fig 4 is to show that with monomeric ALP the sensor is not able to differentiate these concentrations when compared to serum response and their signal almost overlaps the serum signal. Changes have been made to improve the figure for clarity.

Please specify the exact product code (from Biolegend) of the primary antibody used to capture TNFa, and also if it is monoclonal or polyclonal.  

Response: For the study monoclonal antibodies were used and product codes have been included in the revised manuscript.

Round 2

Reviewer 2 Report

The authors have addressed majority of my comments. In Figure 6a, the legend for curve (v) is not consistent with the caption. After they correct this, the paper can be published in my opinion. 

Author Response

Reviewer 2: The authors have addressed majority of my comments. In Figure 6a, the legend for curve (v) is not consistent with the caption. After they correct this, the paper can be published in my opinion. 

Response: Thank you for noticing this error. The legend in Figure 6 has been corrected to “(v) 25 ng/ml TNF-α”.

Reviewer 3 Report

I think that there are still a issue that I feel need addressing.

1) (Response 5) I can’t understand your comment as “PBST 20 SB, which contain various sizes of proteins and effectively block such active COOH sites”. What’s the composition of PBST? Are there various sizes of proteins within PBST? How can PBST block the remaining COOH active sites?

minor points

2) Please add the full spelling for all acronyms of Table 1. For example, DTSP, EA, SAM and QCM.

3) And, according to your comment, TNF-alpha in Ref. 4 (Table 1) seemed to be measured in very diluted serum samples. Please clarify the sample medium in Table 1.

Author Response

Reviewer 3: I think that there are still a issue that I feel need addressing.

1) (Response 5) I can’t understand your comment as “PBST 20 SB, which contain various sizes of proteins and effectively block such active COOH sites”. What’s the composition of PBST? Are there various sizes of proteins within PBST? How can PBST block the remaining COOH active sites?

Response. PBST is a ‘cocktail’ of different proteins with different sizes. Unfortunately we don’t have access to its composition as that is proprietary information from Fisher Scientific. The amine groups of the proteins in PBST will bond to any remaining active COOH groups, furthermore the proteins will also electrostatically couple to any inactive COOH groups.

minor points

2) Please add the full spelling for all acronyms of Table 1. For example, DTSP, EA, SAM and QCM.

Response: We’ve added the definition of the remaining acronyms to the table footer.

3) And, according to your comment, TNF-alpha in Ref. 4 (Table 1) seemed to be measured in very diluted serum samples. Please clarify the sample medium in Table 1.

Response: Although most of the previous electrochemical sensors for TNF-α have been optimised in diluted serum samples, the one in Ref 4 (our previous work) was in undiluted serum as well – and hence the only serum-based comparator included in the Table. We added “undiluted serum” to the respective entries in the table to avoid confusion. The current work, besides the more robust surface chemistry, provides a better linear range of detection.